# Quantitative Study of the Lateral Sealing Ability of Faults Considering the Diagenesis Degree of the Fault Rock: An Example from the Nantun Formation in the Wuerxun-Beier Sag in the Hailar Basin, China

**Xinlei Hu** *[ID], **Yanfang Lv, Yang Liu and Junqiao Liu**

College of Earth Sciences, Northeast Petroleum University, Daqing 163318, China; 571128lyf@nepu.edu.cn (Y.L.); 121991042144@nepu.edu.cn (Y.L.); 121989081906@nepu.edu.cn (J.L.)
* Correspondence: dqsyhuxinlei@nepu.edu.cn

**Abstract:** The goal of this study was to accurately evaluate the lateral sealing ability of a fault in siliciclastic stratum based on previous analysis of the lateral sealing of faults by a large number of scholars in the published literature and physical simulation experiments. Content of the clay mineral phase and the diagenetic degree of fault rock were investigated as the main factors to evaluate the lateral sealing of faults. Based on this theory, the configuration relationship between the clay content and burial depth of fault rock (SGR&H) threshold evaluation method for the lateral sealing of faults was established. Then, we applied these results to evaluate the lateral sealing ability of faults in the Beixi, Beier, Wuerxun, and Surennuoer areas in the Hailar Basin, China. The variation in SGR boundary values with burial depth between the lateral opening and moderate sealing area, as well as between the moderate and strong sealing area of the faults, are obtained. Compared with the previous methods, the SGR&H threshold method transforms the static SGR value of a formation or even a region into a dynamic SGR value that changes with the burial depth, which can fully characterize the differences in the conditions required for sealing faults with different internal structures at different depths. In determining the lateral sealing ability of faults by comparing the evaluation results, we discovered the following. (1) In the same area, the sealing thresholds of faults within different layers are different because the deep strata are subjected to greater pressures and longer loading times, so these faults are more likely to seal laterally. (2) In the same layer, the sealing thresholds of faults in different areas are also different. The higher the thickness ratio between the sandstone and the formation (RSF), the smaller the entry pressure of the fault rock when it has reached a critical seal state, so the SGR&H thresholds are relatively small. Compared to the previous methods, the SGR&H threshold method in this article reduces the exploration risk of faults with relatively low diagenetic degree in shallow strata, and also increases the exploration potential of faults with relatively high diagenetic degree in deep strata. The evaluation results are more consistent with the actual underground situation.

**Keywords:** fault lateral sealing ability; clay content of fault rock; diagenetic degree of fault rock; SGR&H threshold; quantitative analysis; Hailar Basin

## 1. Introduction

Most of the sedimentary basins in China are characterized by thin interbedded sand and clay [1]. The validity of different traps differs even within the same basin because of the tectonic movement and time–space distribution characteristics of sedimentary facies [2,3], especially in the Hailar Basin, which is a small complex rift basin that has undergone several stages of tectonic evolution [4]. Under similar hydrocarbon accumulation conditions (such as source, migration path, etc.), the oil and gas distribution relationship is affected by trap

effectiveness, and the difference in trap effectiveness is largely affected by the difference in lateral sealing ability of faults [3,5].

A fault zone consists of two architectural elements: fault core and fault damage zone [6–9]. The fault core is located in the center of the fault zone and its width can range from millimeters to several meters. It can be composed of a single or multiple fault surfaces and types of cataclastic material, such as breccias and cataclasites, which represent intense deformation [10–14]. The damage zone is distributed symmetrically [15] or asymmetrically [16] on both sides of the fault core. It can be composed of brittle, mechanically related fracture sets, small faults, veins, and joints which represent slight deformation, and the fault rocks retain the basic characteristics of the surrounding rocks [17–20].

According to the relationship of porosity and permeability between fault rock and reservoir rock, the types of lateral fault sealing can be divided into two categories: juxtaposition seals and fault rock seals [21,22]. Juxtaposition seals only occur when (1) the fault scale is relatively small and the fault zone structure is not fully developed [23–25] or (2) the porosity and permeability of fault rock are higher than those of reservoir rock, as the reservoir and the surrounding rocks between the two walls of the faults are juxtaposed and obstruct hydrocarbon migration by the seepage force [26,27]. Otherwise, the sealing type acts as the fault rock seals [28]. The detrital materials in the fault zone that cut from the surrounding rocks gradually discharge pore water and slowly compact into rock due to the influence of diagenesis [29,30], while the porosity and permeability gradually deteriorate. Therefore, the fault rock seals may occur if fault rocks with low permeability and high capillary threshold pressure are generated within the fault zones [31].

With the gradual improvement in fault-related theories such as formation mechanism, internal structure characteristics, sealing mechanism and type of faults, the methods for evaluating the lateral sealing ability of faults have also improved. However, these methods are mostly indirect evaluations of a fault's lateral sealing ability based on the continuity of clay/phyllosilicate smears or of the average clay content within the fault zones, e.g., (1) the clay smear potential (CSP), which is suitable for specific shear environments [32,33]; (2) the shale smear factor (SSF), which is suitable for extrusion environments [34]; and (3) the shale gouge ratio (SGR), which considers multiple geological elements such as the thickness and clay content of strata surrounding the fault throw interval, as well as the displacement of fault [35,36]. In addition, the results calculated by the SGR formula are in accordance with the actual clay content of the fault zone obtained through field calibrations [37]. Geologists prefer to use the SGR formula to evaluate the lateral sealing ability of faults [38,39]. (4) The SGR threshold method judges the lateral sealing ability of a fault with comprehensive consideration of the oil test conclusion, the SGR value of the corresponding section [40–43]. (5) The difference between the entry pressure of the fault rock and the reservoir rock is based on the sealing mechanism of the fault [30,44].

The above methods can determine the lateral sealing ability of fault from a certain point of view, but the CSP, SSF, and SGR formulas only consider the role of the clay content in the fault zone and ignore the diagenesis of the fault rock, which is one of the important factors controlling the lateral sealing ability of the fault. Despite the SGR formula [40] taking account of the effect of compaction diagenetic in fault sealing evaluation, (1) the data involved in the establishment of formula are all come from marine and transitional facies that all over the world (but do not include China), whether it is applicable to continental basins (such as the Hailar Basin), is still uncertain, (2) the burial depth is considered in the form of range (<3.0 km, 3.0–3.5 km, >3.5 km), in the same range the evaluation formula is consistent, but for different areas the sealing properties of faults may vary in the same depth range, which will inevitably lead to misjudgment, (3) in the traditional formula, regardless of the size of the SGR, a certain height of hydrocarbon column can be sealed even if the SGR is 0, so it is inevitable to result in a misjudgment when evaluating open fault. Although the method of

difference between the entry pressure of the fault rock and the reservoir rock is more comprehensive, it is necessary to measure the value of the entry pressure of the rock samples from the target area in the laboratory, which is rather limiting and is not conducive to the rapid evaluation of the lateral sealing ability of a fault.

Because of these factors, this paper establishes a set of SGR&H threshold methods to quantitatively analyze the lateral sealing ability of the fault that consider both of the influential factors proposed by previous studies and is extremely suitable for areas with complex structural units such as depressions, slopes, and uplifts (such as the Hailar Basin). The SGR&H threshold method can distinguish the difference between the lateral sealing ability of faults in deep and shallow strata and deduce the sealing threshold of faults in less developed strata so that the threshold values of multiple sets of strata are continuous. Compared with previous research methods, the SGR&H threshold method established in this article has certain advantages. It not only quantitatively considers the influence of two parameters, fault clay content and diagenetic degree, on the lateral sealing of faults but also obtains the configuration relationship between these two parameters by establishing template. Generally, in shallow strata, the diagenetic degree of rock is low, and it is necessary to configure faults with high clay content to form a seal, while in deep strata, the diagenetic degree of rock is high, and it is only necessary to configure faults with low clay content to form a seal. In other words, there is a depth demarcation point. When the actual depth is less than the critical depth, the threshold of lateral fault sealing obtained by the SGR&H method is greater than that of the SGR method, resulting in the actual opening fault being evaluated as a sealing fault. This difference can be used to explain the contradiction of using the SGR method to drill into aqueous layers in the fault sealing area of shallow strata. However, when the fault depth is greater than the critical depth, the threshold of lateral fault sealing obtained by the SGR&H method is smaller than that of the SGR method, resulting in the actual sealing fault being evaluated as an opening fault. This difference can be used to explain the contradiction of using the SGR method to drill into hydrocarbon layers in the fault opening area of deep strata.

Therefore, this research is of significance for guiding the fast and accurate evaluation of the lateral sealing ability of faults in mature exploration areas, reducing the risk of drilling in fault-related traps, and improving our understanding of oil and gas accumulation in fault areas, which will be valuable for efforts in identifying new target areas.

## 2. Geological Setting

The Hailar Basin is located in the Inner Mongolia Orogenic Belt between the Siberian Plate and the North China Plate [45]. The northern part of the basin is bordered by the Potalaira Basin, the southern part is bordered by the Tamtsag Basin in Mongolia, and it is clamped by Banyan Mountain and the Cuogang Uplift in the east and west, respectively. The Hailar Basin is a Mesozoic–Cenozoic continental rifted basin superimposed on the Mongolia–Hinggan orogenic belt [4,46]. It is composed of two parts: the Wuerxun Sag in the north and the Beier Sag in the south. The former includes the Surennuoer and Wuerxun areas, while the latter includes the Beixi and Beier areas (Figure 1).

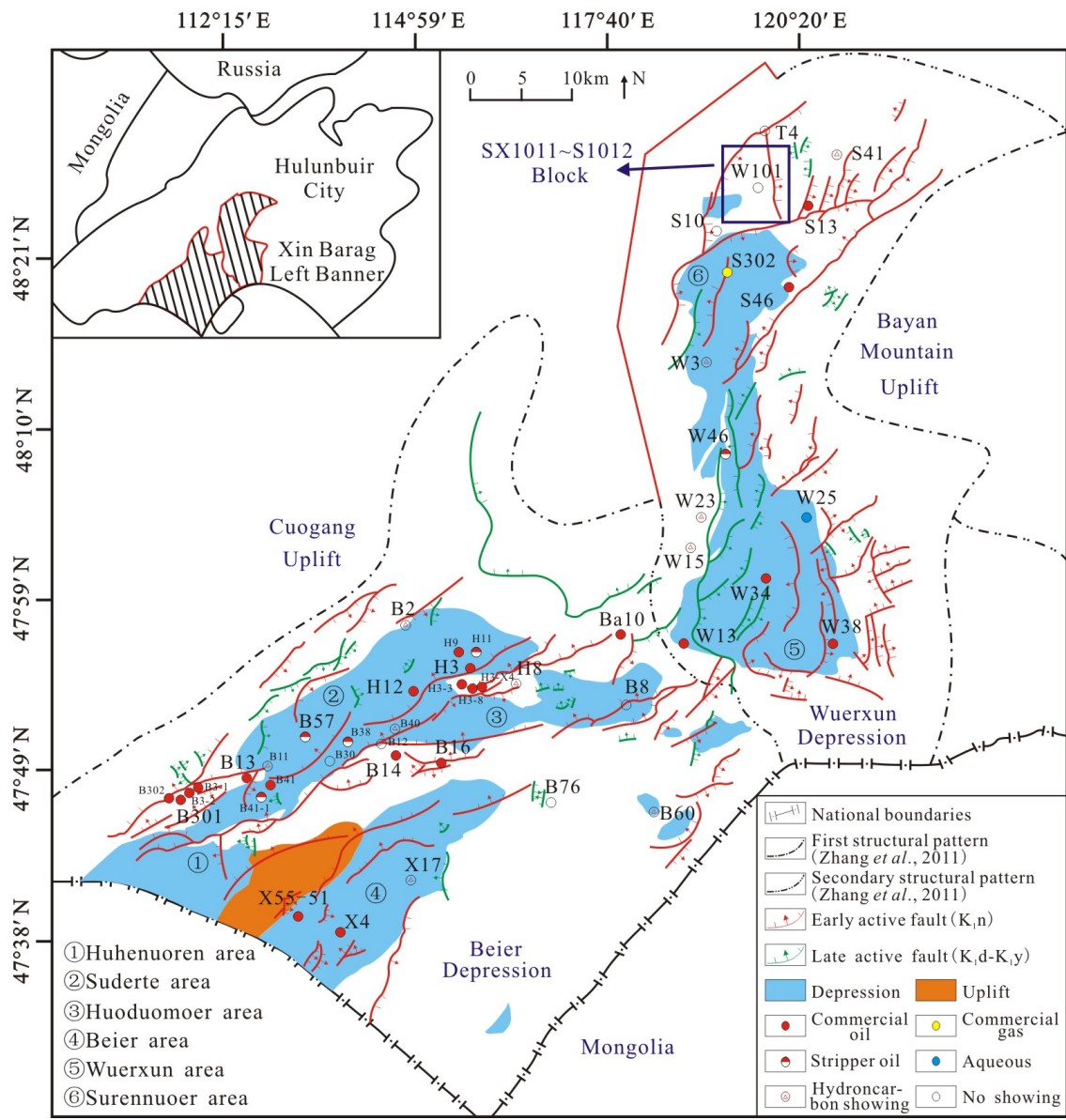

**Figure 1.** Regional geological map of the Hailar Basin. (Revised by [47]).

The basement of the Hailar Basin is composed of Hercynian–Indosinian metamorphic rock and the Budate Group [48,49]. Its interior is composed from bottom to top of the Cretaceous and Cenozoic strata of the Tongbomiao ($K_1t$), Nantun ($K_1n$), Damoguaihe ($K_1d$), Yimin ($K_1y$), and Qingyuangang ($K_2q$) formations (Figure 2). The characteristics of each formation are as follows. (1) The $K_1t$ Formation is angular unconformity above the basement, it is the sedimentary filling product of the residual basin in the early Cretaceous, whose thickness is about 250–700 m (820–2297 ft). The lower part of this formation is a set of normal-cycle sediments composed of andesite, tuff and tuffaceous sand–mudstone, while the upper part is a set of inverse-cycle sediments of glutenite. (2) The $K_1n$ Formation has a great difference in lithology in different parts of the basin. The formation in the central part is mainly composed of dark-gray mudstone and argillaceous siltstone, which are high-quality source rocks, while the grain size in the marginal area of both sides of the basin gradually becomes coarser, which is mainly composed of conglomerate, pebbly coarse sandstone and other coarse clasts, which are high-quality reservoirs. (3) The $K_1d$ Formation is mainly composed of gray-black mudstone, black mudstone and silty mudstone. It is a set of normal-cycle sediments with a thickness of 350–600 m (1148–1969 ft), and the

giant thick mudstone developed at the top can serve as a regional seal for the whole basin. (4) The $K_1y$ Formation is composed of 600–1000 m (1969–3281 ft)-thick green, gray siltstone, sandstone and mudstone with a small amount of coal seam. (5) The $K_2q$ Formation is mainly composed of <500 m-thick pink, red silty mudstone, and muddy siltstone interbedded with sandy conglomerate.

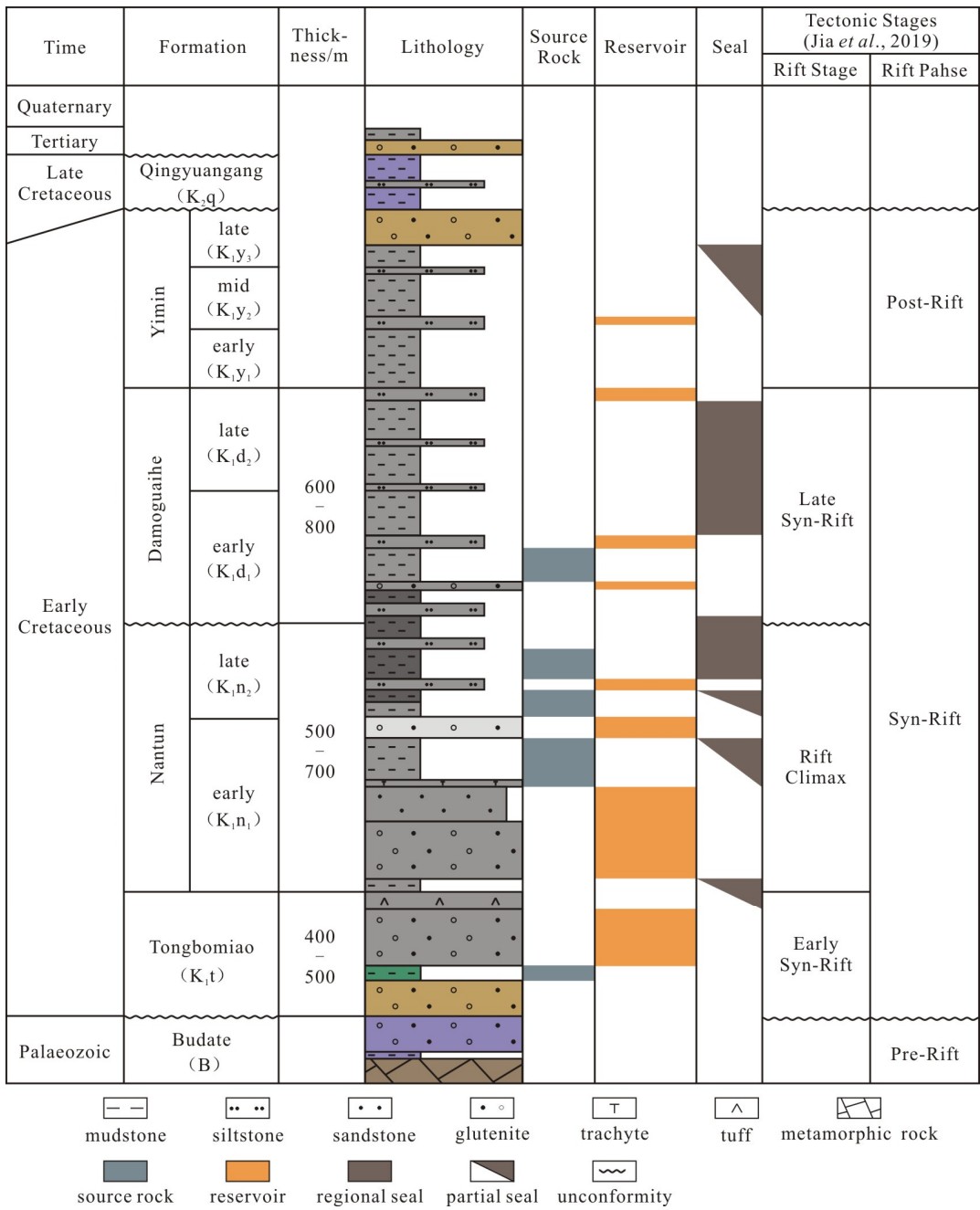

**Figure 2.** Comprehensive stratigraphic map of the Hailar Basin. (Revised by [50]).

This basin has experienced five periods of evolution: the remnant basin period, the initial rifting period, the intense rifting period, the rift-sag period, and the post-rifting period [50]. Due to multiple periods of construction and reconstruction, the Wuerxun-Beier Sag in the Hailar Basin formed a complex fault system, which is characterized by multiple properties, multiple strike directions, multiple combinations, and multiple periods. During the deposition period of the $K_1t$ Formation, the faults began to form, but the size of faults and the throws were not large, which formed an obvious uplift and erosion. Subsequently,

during the deposition period of the $K_1n$ Formation, the basin entered the intense rifting period, while with the strong fault activity, fault size and throw significantly increased, and most of the faults have a NNE–NE strike due to the SE–NW tensile stress [2] and the structure of the preexisting basement fault block [46,51]. These faults are mainly shown as listric faults in cross section, as well as echelon arrangement in the plane. During the deposition periods of the $K_1d$ to $K_1y$ Formations, the early SE–NW tensional effect transitioned to a nearly EW direction, so the strike of the faults that formed in this stage is NS and NNW. These faults mainly have flower-like or y-shaped features in cross section, and pectination or fish ridge combination in the plane. Finally, the basin was controlled by left-axis compression and torsion that caused a strong reversal of the faults generated in the early stages. Hydrocarbons have mainly been discovered in fault-block traps and fault-related anticlines, strata and lithologic traps. Thus, whether the faults have lateral sealing ability is the key to determining the validity of fault-related traps and the reasons for hydrocarbon accumulation in the Hailar Basin.

The main focus of this study is the $K_1n$ Formation, which is composed of a relatively complete depositional system including lowstand, lacustrine and highstand system tracts [52,53], and it is the main source rock as well as the major reservoir in the study area. For the early extensional faults formed during the deposition period of the $K_1n$ Formation [2] in particular, whether these faults have lateral sealing ability is the key to hydrocarbon accumulation.

## 3. Quantitative Analysis Method and Technique

### 3.1. Factors Influencing the Lateral Sealing Ability of the Faults

Based on the lateral sealing types of faults and the results of previous studies, we investigate and confirm that the clay content, the degree of diagenesis of the fault rock, and the history of fault activity are the three main geological parameters that control the lateral sealing ability of faults. Other factors can be accounted for indirectly by using these three parameters.

First, the lateral sealing ability of a fault is proportional to the clay content of the fault rock (SGR). The higher the SGR, the greater the threshold pressure required for lateral migration of hydrocarbons across the fault [35,54,55]. According to the clay content in the fault zone, four cases can occur [56]. (1) For pure sandstone with a clay content less than 15%, disaggregation zones form due to the low effective stress (mechanical compaction) without a significant decrease in the porosity and permeability, and the lateral sealing ability of the fault rock is relatively weak. (2) Pure sandstone forms cataclasite due to a high effective stress (cataclasis) or cementation; thus, the lateral sealing ability of the fault is relatively strong [57]. (3) Impure sandstones containing 15–40% clay form phyllosilicate-framework fault rocks due to shear stress, quartz cementation, and pressure solution [58], which results in a relatively strong fault sealing ability. (4) Impure sandstone with a clay content greater than 40% develops clay smearing due to the effective stress, which results in a strong sealing ability of the fault [59,60].

With increasing degree of diagenesis, the fault rock's porosity and permeability gradually decrease, and thus the faults are more likely to form lateral seals [29]. During the burial process, with the increase in burial depth and underground temperature, the fillings in the fault zone during diagenesis are: (1) influenced by mechanical compaction—the pore water is gradually discharged under the effect of the overlying load and regional principal stress, and then slowly compacted; (2) influenced by chemical cementation, minerals in fillings from supersaturated precipitation sealing the fractures or even pores; (3) influenced by dissolution—secondary porosity can be formed, but the modification degree of the fault rock by the dissolution is obviously weaker than that of the mechanical compaction and chemical cementation. (4) In addition, according to the fault deformation mechanism, the fault rock may successively undergo disaggregation, cataclasis and shear smear in the process of deep-buried diagenesis, and gradually transform from breccia to finer clastic particles and argillaceous. Thus, with the increase in burial depth, the permeability of

the fault rock decreases gradually [61]. This is also characterized by the increase in the diagenesis degree of fault rock and the formation of sealing ability of faults (Figure 3).

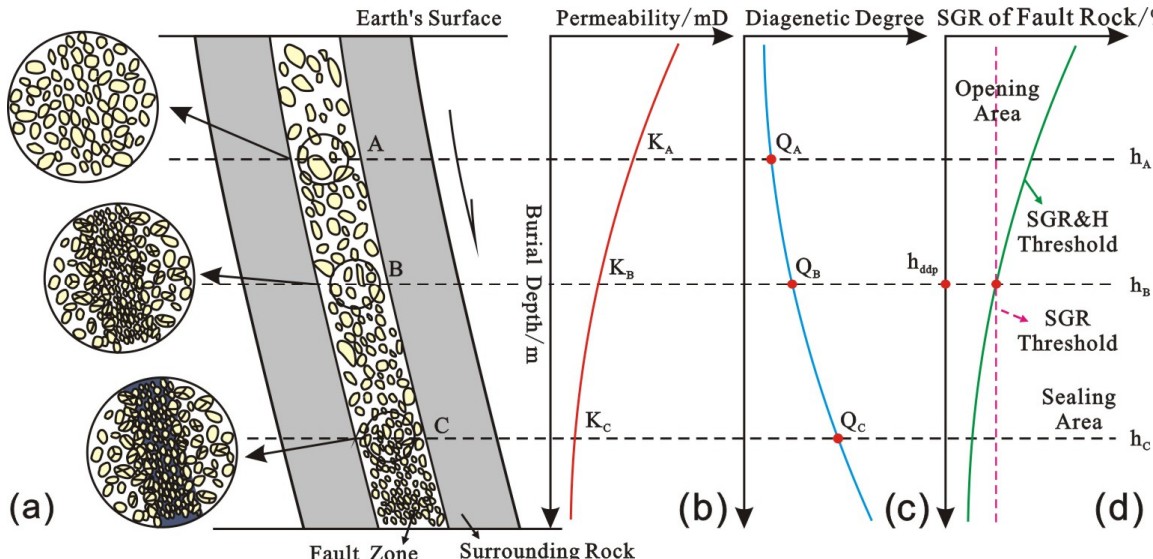

**Figure 3.** Relationship between internal structure of fault zone (**a**), permeability (**b**), diagenetic degree (**c**) and sealing threshold of fault rock (**d**) with burial depth, where points *A*, *B* and *C* are typical points of faults, $K_A$, $K_B$ and $K_C$ are the permeability of each points, $Q_A$, $Q_B$ and $Q_C$ are the diagenetic degree of each points, $h_A$, $h_B$ and $h_C$ are the burial depth of points *A*, *B* and *C*, $h_{ddp}$ is the burial depth of depth demarcation point.

It should be noted that considering different faults in different sedimentary environments or even different parts of the same fault are affected by the difference of fracture development degree and hydrothermal subsurface, the fillings in the fault zone have certain diagenetic heterogeneity in the longitudinal and lateral directions, so it is often inaccurate, unrepresentative and ungeneralized to characterize the diagenetic characteristics of fault rocks by using limited data of coring wells drilled in the fault zone. Because the fault rocks are more prone to cementation than surrounding rocks, this text only uses burial depth to represent the minimum diagenesis degree of fault rocks, and the lateral sealing ability of fault based on this criterion is also the safest and most reliable.

The relationship between the activity history of the fault and the hydrocarbon accumulation period also controls the fault sealing. If the sealed faults that formed in the early stages of deposition are not active during the period of hydrocarbon accumulation, they would capture large amounts of oil and gas. Conversely, if the faults reactivate, they may lose their lateral sealing ability [62], and the oil and gas that accumulated earlier will be released. Therefore, the analysis of fault activity should be carried out according to the actual situation of the study area. If the active period is earlier than the hydrocarbon accumulation period, it has no significant effect on the lateral sealing of faults. If the active period coincides with or is later than the accumulation period, it is necessary to analyze the destroy of the already formed accumulation by fault activity.

### 3.2. Determining the SGR&H Threshold of Lateral Fault Sealing

According to the above analysis, clay content and degree of diagenesis of the fault rock are the two major geological factors controlling the lateral sealing ability of faults, while the effect of the fault activity history needs to be analyzed in detail according to the specific situation, which will not be elaborated upon in this part. Therefore, an SGR&H threshold method for quantitatively analyzing the lateral sealing ability of faults is proposed, which considers the influence of the clay content and differences in the sealing ability at different burial depth. This method is suitable for blocks with weak fault activity after the hydro-

carbon accumulation period, which is of great significance to hydrocarbon exploration in mature blocks. The procedures for this method are as follows.

(1)   Determination of the SGR&H of the fault rock

The SGR formula [35,36] uses the average value of the clay content of the beds that have slipped past the target point (as determined by the vertical fault throw) as an estimate of the upper limit of fault-zone composition [63]. Therefore, the model can use the seismic interpretation results for the study area to establish a three-dimensional structural model of the formations and faults and to calculate the vertical throw of the target faults at different depth. Then, we use the logging data (SP, GR, or other curves) to determine the variation in clay content of the surrounding rocks. Then, we can obtain the SGR value of the fault rocks at different burial depths using Equation (1) [35]; that is to say, combined with the corresponding buried depth (H), the SGR&H value of any points along fault section can be determined (Figure 4a):

$$SGR = \frac{\sum\limits_{i=1}^{n} \Delta Z_i \cdot V_{shi}}{L} \times 100\% \tag{1}$$

where SGR is the clay content of the target fault rock (%); n is the number of sand and clay beds that slip past the target fault rock; $\triangle Z_i$ is the thickness of bed i that slips past the target point (m); $V_{shi}$ is the clay content of bed i that slips past the target point (%); and L is the vertical throw of the fault (m).

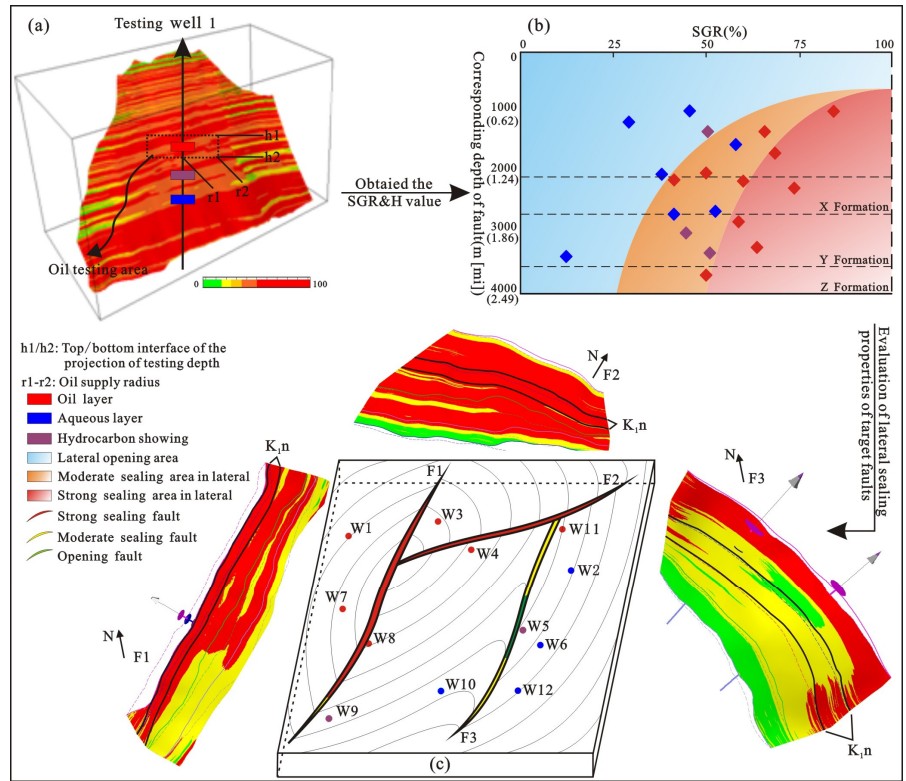

**Figure 4.** Flow diagram of the determination of the SGR&H threshold. (**a**) SGR&H attributes of the fault rock. (**b**) Evaluation template and SGR&H threshold for fault sealing. (**c**) Determination of faults' lateral sealing properties.

(2)   Constraining the oil testing area

Considering the influence of the dip angles of the fault and reservoir, the actual oil testing depth within the well is projected onto the corresponding fault section, and the top and bottom interfaces (h1–h2) control the longitudinal range of the oil testing area,

while the oil supply radius of the target well (r1–r2) and the high position of the fault on an upward inclination jointly control the lateral range of the oil testing area [64,65].

(3)　Determination of the SGR&H threshold for lateral fault sealing

For establishment of the evaluation model, first, we selected the typical wells within the target area, which should follow the two descriptions to effectively avoid the effects of hydrocarbon migration heterogeneity or other geological factors: The first is that the wells are located in the fault-related traps, rather than the anticline or lithologic traps, only in this case, the preservation and sealing conditions of the selected wells are the faults, and only the lateral sealing faults are the necessary conditions for oil and gas accumulation. The second is that the oil supplement, reservoir quality, distance from migration path and other accumulation conditions are well matched (all of them should reach the threshold of oil and gas accumulation). At this time, whether the typical well can discover oil and gas is completely affected by the sealing conditions, which are affected by the lateral sealing property and ability of faults. If the fault is sealed, oil and gas will be discovered above the oil–aqueous interface corresponding to the minimum SGR value of the fault rock, and vice versa, oil and gas showing or aqueous layer will be discovered. Then, we determined the oil testing area of the different layers at different depths above the wells on the corresponding fault surface and the minimum SGR value in the oil testing area. Taking the minimum SGR value of the fault surface in the oil testing area as the horizontal coordinate and the oil testing depth as the longitudinal coordinate, we constructed a scatter-point map. Based on the oil testing results of the different wells, the boundaries of the lateral opening area and the moderate and strong sealing area were determined, and an evaluation model of the lateral fault sealing ability was established (Figure 4b). In this model, the blue area represents the lateral opening area of the fault, for which the wells controlled by them are all aqueous layers. The yellow area represents the moderate sealing area of the fault, for which the wells controlled by them can be either aqueous layers or commercial oil layers, and the results depend on the height of hydrocarbon column calculated by SGR and the oil testing depth. The red area represents the strong sealing area of the fault, for which the wells controlled by them are all commercial oil layers.

Determination of the SGR&H threshold: According to the distribution characteristics of the lateral opening area, moderate sealing area and strong sealing area that divided in the previous procedures, the relationship between the SGR value and the burial depth of fault rock under different critical conditions can be determined using Equations (2) and (3). The fitting relationship between the lateral opening area and the moderate sealing area is shown as the SGR&H threshold for fault sealing:

$$SGR_{O-MS} = f(H) \tag{2}$$

$$SGR_{MS-SS} = f(H) \tag{3}$$

where $SGR_{O-MS}$ is the SGR value of fault rock at the critical condition between lateral opening and moderate sealing stage when the burial depth is equal to H (%); $SGR_{MS-SS}$ is the SGR value of fault rock at the critical condition between moderate sealing and strong sealing stage when the burial depth is equal to H (%); and H is the burial depth of the target fault rock (m).

(4)　Determination of the lateral fault sealing properties

We used the defined threshold as a criterion to judge the lateral sealing properties of different faults in different layers; that is to say, when the location of the minimum SGR value of the target fault and the corresponding burial depth are projected onto the lateral opening area, the fault is laterally open. When the projected point is located in the moderate sealing area, the lateral fault seal and the sealing ability are intermediate. When the projected point is located in the strong sealing area, the lateral fault seal and the sealing ability are strong (Figure 4c).

## 4. Quantitative Analysis of the Lateral Sealing Ability of Faults in the Hailar Basin

### 4.1. Fault Characteristics and Method Applicability

The differences in fault characteristics determine the factors and analysis methods of lateral sealing. In order to better quantitatively analyze the lateral sealing ability of faults in the Hailar Basin, it is crucial to determine the sealing type and whether the SGR&H threshold method established above is applicable.

Therefore, based on the above methods and techniques, we selected a typical geological section near Miandu-Zhadun River of the Hailar Basin, and analyzed the characteristics of the faults and formations in detail. The results in Figure 5a show that the fault in $J_2n$ Formation (which was shallower than the target $K_1n$ Formation) has a certain scale (displacement = 0.4 m) and the structure of fault zone is fully developed. From the actual data of Hailar Basin, the porosity of this fault core (fault gouge) is 2% to 10%, the permeability is 0.05 mD to 0.7 mD, the porosity of reservoir juxtaposed to the fault is 18% to 32%, and the permeability is between 26 mD to 48 mD. The results show that the petrophysical properties of the fault rock are obviously worse than those of the reservoir rock, which indicated that the fault can form a fault rock seal.

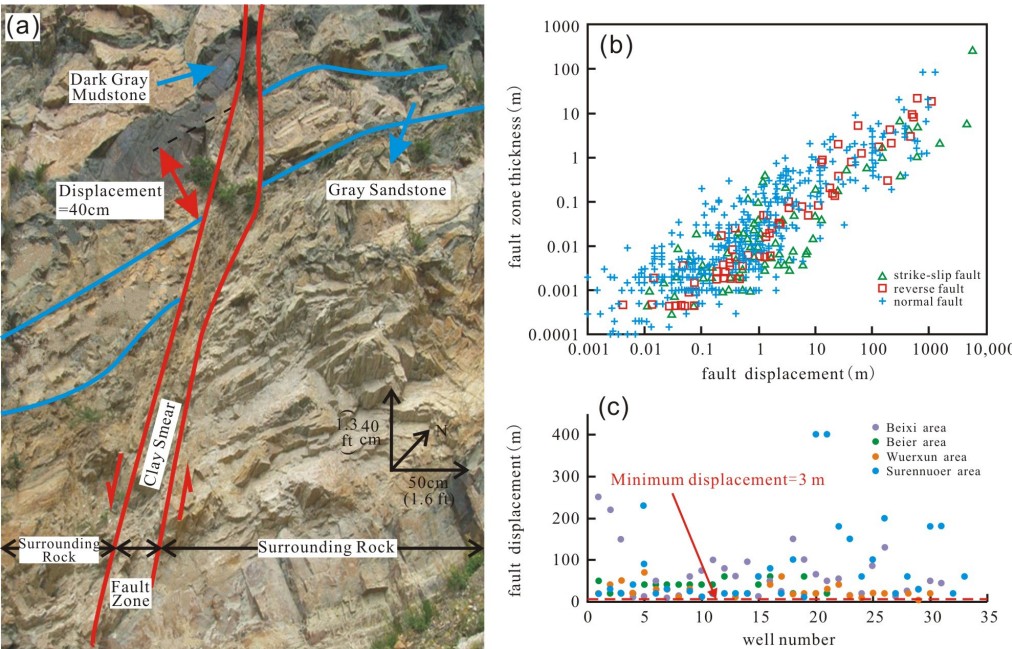

**Figure 5.** (**a**) Typical geological section of faults in $J_2n$ Formation near Miandu-Zhadun River of the Hailar Basin. (**b**) Plot of fault rock thickness versus displacement for the different modes of faulting [7]. (**c**) Fault displacement of typical wells in $K_1n$ Formation of different areas of Hailar Basin.

Fault rock thickness and displacement data from a wide variety of sources revealed that there is a positive relationship between them (Figure 5b): with the increase in fault displacement, the thicker the fault zone thickness is, the higher the grinding deformation degree of the fault rock, the worse its porosity and permeability, and the more easily fault rock seals are formed [7,66]. Thus, since the fault shown in Figure 5a shows a fault zone with a certain width composed of fault rocks when the fault displacement is only 0.4 m. It can be considered that the faults in different areas of the Hailar Basin with relatively large displacement (3–400 m, Figure 5c) are all laterally sealed by fault rock seals.

By analyzing the history of the tectonic evolution of the Hailar Basin, we confirmed that the movements during the late depositional period of the $K_1t$, $K_1n$, and $K_1y$ Formations had a significant effect on the lateral sealing ability of faults [67]. The study area experienced two main stages of hydrocarbon accumulation, which occurred during deposition of the $K_1y$ Formation and from deposition of the $K_2q$ Formation to the present. In particular, the second accumulation stage played an important role in the formation of oil and gas

reservoirs in the Hailar Basin [45,50,68–70]. In contrast, the early extensional faults in the K$_1$n Formation were inactive during the first stage of hydrocarbon accumulation, so the controlling effect of the fault activity history on fault sealing is not obvious in Hailar Basin. In other words, the SGR&H threshold method established above is suitable for quantitative analysis of lateral fault sealing in the Hailar Basin, and can be used for detailed analysis and demonstration.

### 4.2. Quantitative Analysis of the Lateral Sealing Ability of Faults

Based on the theory and method for determining the SGR&H threshold of lateral fault sealing described above, the relationship between the minimum SGR of the fault rock, the oil testing depth, and the testing results of target layers in different oil fields of the Beixi area were obtained. Then, we established the evaluation model (Figure 6a) and determined the SGR&H thresholds as well as curves that divide lateral opening and moderate and strong sealing areas in the Beixi areas (Equations (4) and (5)). Therefore, in the actual evaluation process, the lateral sealing ability of faults can be determined by comparing the actual SGR value of fault rock with the critical SGR$_{\text{O-MS}}$ and SGR$_{\text{MS-SS}}$ value required for lateral opening—moderate sealing and moderate sealing—and strong sealing at corresponding depth. If SGR < SGR$_{\text{O-MS}}$, the faults are laterally opened and do not have the ability to seal hydrocarbon; if SGR$_{\text{O-MS}}$ ≤ SGR < SGR$_{\text{MS-SS}}$, the faults are laterally sealed and have medium sealing ability; if SGR ≥ SGR$_{\text{MS-SS}}$, the faults are laterally sealed and have strong sealing ability, the heights of hydrocarbon column are relatively high. According to the evaluation results of the Beixi area, it can be seen that the main faults in the K$_1$n Formation have moderate to strong sealing abilities that are beneficial to sealing hydrocarbon, and only a small number of opening faults are developed in the center of the Huoduomoer area (Table 1).

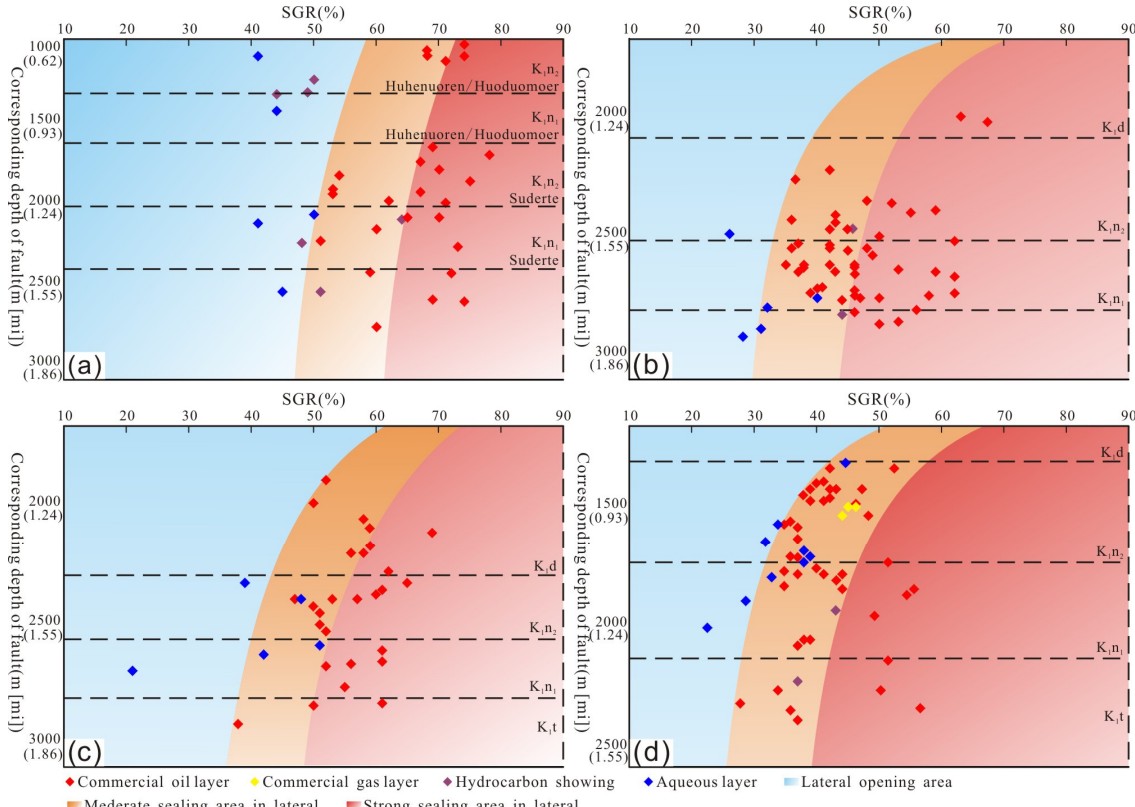

**Figure 6.** Determination of the SGR&H threshold of the faults in different areas and layers of the Hailar Basin. (**a**) Beixi area, which includes the Huhenuoren, Suderte, and Huoduomoer blocks. (**b**) Beier area. (**c**) Wuerxun area. (**d**) Surennuoer area.

**Table 1.** Statistics for oil testing, SGR&H, and the fault sealing attributes of typical wells in the Hailar Basin.

| Area | Well | Testing Layer | Testing Conclusion | Daily Oil Production (Layer Number) (t) (Layer) | Fault Displacement (m [ft]) | Buried Depth (m [ft]) | SGR (%) | Fault Sealing Property |
|------|------|--------------|-------------------|-------------------------------------------------|-----------------------------|-----------------------|---------|------------------------|
| Beixi Area | B3 | $K_1n_2$ | Oil | 6.97 (5) | 15 (49) | 1130 (3707) | 71 | Strong Seal |
| | H3-12 | $K_1n_2$ | Showing | / | 23 (75) | 1245 (4085) | 50 | Open |
| | D124-137 | $K_1n_1$ | Oil | 0.31 (1) | 175 (574) | 1790 (5873) | 62 | Moderate seal |
| Beier Area | X03-61 | $K_1n_2$ | Water | 0 (1) | 12 (39) | 2507 (8225) | 26 | Open |
| | X2-1 | $K_1n_2$ | Oil | 8.48 (1) | 8 (26) | 2440 (8005) | 43 | Moderate seal |
| | X5 | $K_1n_1$ | Oil | 8.64 (3) | 3 (10) | 2696 (8845) | 62 | Strong Seal |
| | X09-55 | $K_1n_1$ | Showing | 0.13 (3) | 13 (43) | 2599 (8527) | 27 | Open |
| Wuerxun Area | W112-88 | $K_1n_1$ | Oil | 1.27 (1) | 25 (82) | 2675 (8776) | 56 | Strong Seal |
| | W144-108 | $K_1n_1$ | Water | 0 (3) | 5 (16) | 2033 (6670) | 40 | Open |
| | W148-70 | $K_1n_1$ | Water | 0 (4) | 18 (59) | 2713 (8901) | 21 | Open |
| | W29 | $K_1n_1$ | Oil | 2.09 (4) | 23 (75) | 2489 (8166) | 53 | Moderate seal |
| Surennuoer Area | S31 | $K_1n_2$ | Oil | 0.21 (1) | 48 (157) | 1545 (5069) | 47 | Moderate seal |
| | XW1 | $K_1n_2$ | Water | 0.06 (4) | 60 (197) | 1601 (5253) | 33 | Open |
| | S20 | $K_1n_1$ | Oil | 4.37 (3) | 9 (30) | 2065 (6775) | 38 | Moderate seal |
| | S15 | $K_1n_1$ | Water | 0 (2) | 32 (105) | 1911 (6270) | 28 | Open |

Using the same method to determine the SGR&H threshold of fault sealing in the $K_1n$ Formation of the Beier, Wuerxun, and Surennuoer areas (Figure 6), we delineated the boundaries between lateral opening, moderate sealing, and strong sealing areas, and then established the corresponding functions, as shown in Equations (6) and (11).

$$\text{Beixi Area}: \text{SGR}_{O-MS} = \frac{Ln\left(\frac{H}{4.10\times10^5}\right)}{-10.3} \tag{4}$$

$$\text{SGR}_{MS-SS} = \frac{Ln\left(\frac{H}{1.83\times10^6}\right)}{-10.3} \tag{5}$$

$$\text{Beier Area}: \text{SGR}_{O-MS} = \frac{Ln\left(\frac{H}{1.12\times10^4}\right)}{-4.45} \tag{6}$$

$$\text{SGR}_{MS-SS} = \frac{Ln\left(\frac{H}{2.04\times10^4}\right)}{-4.45} \tag{7}$$

$$\text{Wuerxun Area}: \text{SGR}_{O-MS} = \frac{Ln\left(\frac{H}{7.25\times10^3}\right)}{-2.57} \tag{8}$$

$$\text{SGR}_{MS-SS} = \frac{Ln\left(\frac{H}{9.99\times10^3}\right)}{-2.57} \tag{9}$$

$$\text{Surennuoer Area}: \text{SGR}_{O-MS} = \frac{Ln\left(\frac{H}{6.54\times10^3}\right)}{-3.98} \tag{10}$$

$$\text{SGR}_{MS-SS} = \frac{Ln\left(\frac{H}{1.08\times10^4}\right)}{-3.98} \tag{11}$$

Through the analysis of Figure 6 and Equations (4)–(11), it can be concluded that in the shallow strata, the fault rock is subjected to small lithostatic pressure, which results in a relatively low degree of diagenesis. Therefore, the fault rock needs to reach a higher SGR to form a lateral seal. Thus, there exists an upper depth threshold. When the fault is at this depth and the fault rock behaves as pure shale (SGR value equal to 100%), it reaches

the critical transition state of an open seal. With increasing burial depth, the pressure on the fault surface gradually increases as the overlying sedimentary load and the degree of diagenesis of fault rock increases. The SGR threshold of lateral fault sealing decreases correspondingly. When the SGR of the fault rock reaches a certain value, with increasing burial depth, changes in the sealing threshold are not obvious. The relationship between the above influencing factors controls the boundaries of the fault opening–sealing area, which appears to be upper gentle and lower steep (Figure 5b). At the same depth, the higher the SGR of the fault rock is, the easier it is to form a lateral seal. Only when the SGR is greater than or equal to the threshold of lateral fault sealing does the fault have the ability to seal oil and gas laterally. For the same SGR, the sealing properties of the fault rock at different depths are different. The larger the breakpoint depth is, the smaller the threshold required for lateral fault sealing.

In summary, compared with the previous research methods, the method established in this paper considers the main controlling factors of fault lateral sealing more comprehensively and strengthens the connection and restriction between multiple main controlling factors through the establishment of the relationship between the burial depth, clay content of fault rock and testing results of typical wells. To a certain extent, it can reflect the control effect of the internal structure and porosity of the fault zone at different depths on the properties and sealing ability of fault. However, the previous methods have considered the diagenetic degree of fault rocks too roughly or even not at all, nor have they considered the relationship between the clay content and diagenetic degree. A change in one factor controls the value of the other factor. Therefore, the evaluation results are not more consistent with the actual underground conditions than those obtained by the SGR&H method.

### 4.3. Analysis of Typical Cases

The B3 Fault is located in the Huhenuoren Oilfield, its displacement in $K_1n_2$ is equal to 80 m (262 ft), the structure of fault zone is fully developed and its porosity (5.5–8.3%) is lower than the surrounding rocks (10.5–34.3%), which indicates that the B3 Fault forms a fault rock seal. In this case, whether the fault sealing is controlled by the fault rock rather than the juxtaposed clay from the $K_1d_1$, the sealing ability is affected by the clay content and the burial depth of the fault rock. Then, we analyzed the SGR properties of the target fault (Figure 7a) and found that the SGR values of the fault rocks at the top of $K_1n_2$ Formation are relatively large. The minimum SGR value (69%) is greater than the $SGR_{MS-SS}$ threshold (58%) at corresponding burial depth, which indicates a strongly sealed fault. With increasing burial depth, the surrounding rocks of the fault gradually transition from a large set of mudstone to interbedded sand and mudstone (Figure 7b). Because the fault rock is formed by surrounding detritus, which is cut and falls into the fault zone when the fault slides, the decrease in the clay content on both sides of the fault controls the decrease in the SGR of the fault rock. Within the same depth range in $K_1n_2$, the lateral fault sealing ability is weakened, and it is gradually transformed into a moderately sealed fault. Because the B3 Fault has a certain lateral sealing ability in $K_1n_2$, due to the effects of the reservoir quality (porosity of 21.6%, permeability of 18.86 mD) and hydrocarbon migration, the testing results for the $K_1n$ Formation at the B3 Fault show a transition from a commercial oil layer to a hydrous oil layer, and eventually to an aqueous layer (Figure 7c). Similarly, the differences in the SGR&H values of the fault rock in the plane also control the distribution of the different reserves. Areas B301 and B13 have relatively high SGR&H values and are expressed as proven reserves, while area B70 is a prognostic reserve with a lower SGR&H value than that of the first two areas, but is still controlled by sealing faults. Therefore, moderate to strong sealing faults are favorable faults for sealing oil and gas. The stronger the lateral sealing ability is, the more advantageous the fault is to oil and gas accumulation.

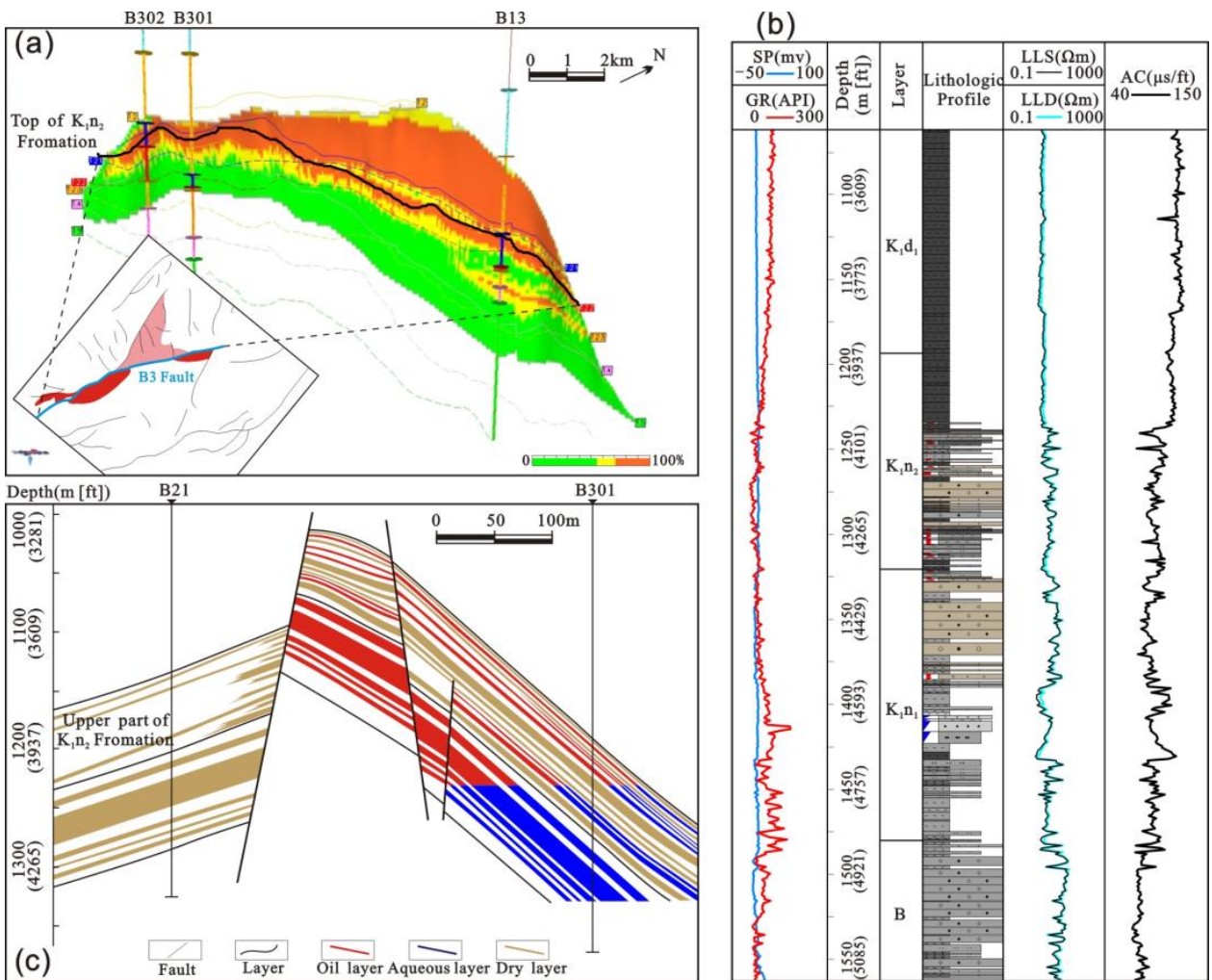

**Figure 7.** Map of the lateral sealing ability of the B3 Fault in the Hailar Basin. (**a**) SGR value of the B3 Fault. (**b**) Composite columnar section of well B301. (**c**) Reservoir profile across well B301.

For area B70 in the northern Huhenuoren Oilfield, the porosity of the B70 Fault (2–5%) is smaller than that of the surrounding rocks (6–23%); thus, the sealing type of target fault is also fault rock seal. The lateral sealing property of the fault in the $K_1n_2$ Formation is mainly influenced by the SGR value and diagenetic degree of the fault rock. Figure 8a illustrates the fact that the sealing properties are different along different positions of the B70 Fault. Combined with the corresponding burial depth, it can be seen that the upper part of the fault acts as a moderate sealing fault, while the lower part is laterally open by comparing the SGR value with the $SGR_{O-MS}$ and $SGR_{MS-SS}$ threshold. Due to the fact that the tail of the B70 Fault lacks lateral sealing ability, the effective scope of the target trap is reduced on the basis of the original trap. When the other reservoir forming conditions are favorable, successful wells are drilled within the scope of the effective trap, such as wells B17 and B70, which intersect oil layers (Figure 8b), while failed wells are drilled outside the effective trap or even outside the scope of the trap, such as wells B17-110-42, which lack hydrocarbons. Hence, opening faults may reduce the effective scope of a trap and even lead to failure of the entire trap, and the development of an abundance opening faults in the northern part of the Wuerxun area and southwestern part of the Surennuoer area is the main reason for the abundance of failed wells in these areas.

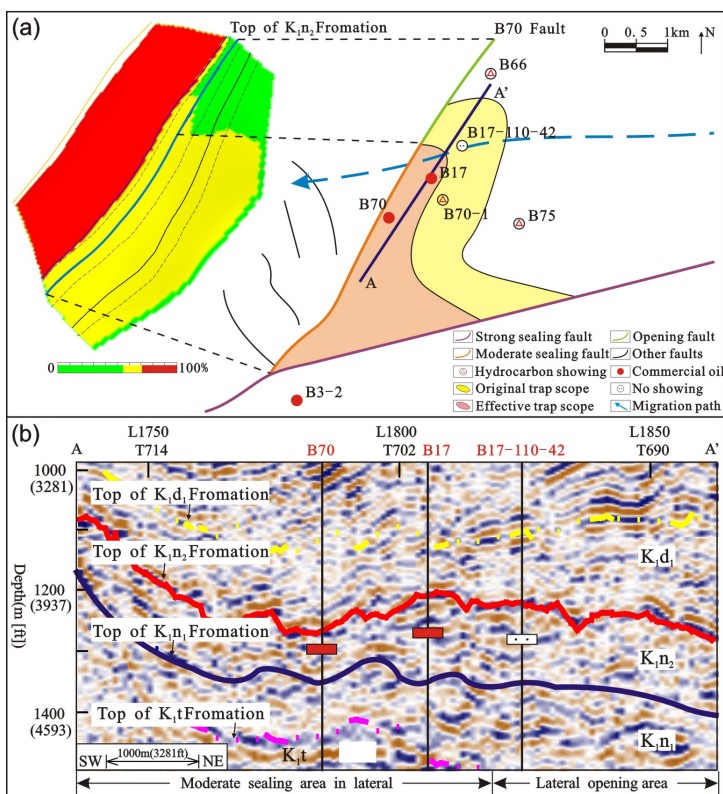

**Figure 8.** Map of the lateral sealing ability of the B70 Fault in the Hailar Basin. (**a**) SGR attributes of the B70 Fault and the relationship between the hydrocarbon discovery with the effective/original trap scope. (**b**) Seismic profile AA' shown in Figure 7a.

Compared the evaluation results with earlier calibration of SGR against the burial depth [63,71], no matter what the SGR of the fault rock is, the trap can seal a certain hydrocarbon column, while for the method in this text, if the SGR&H is lower than the threshold, the trap is invalid and has no sealing ability, which indicates that the former method might overestimate the sealing ability of faults.

### 4.4. Analysis of Application Effect

The above analysis confirms that the evaluation results of the lateral fault sealing ability obtained using the method established in this text are consistent with the actual oil–water distribution in the study area. This method has been widely used to evaluate the fault sealing ability and to identify new targets in the $K_1n$ Formation in the immature blocks in the Hailar Basin.

Block SX1011-S1012 is located in the northern part of the Surennuoer area (Figures 1 and 9a), and the troughs on the eastern and southern sides of this block have a certain hydrocarbon-generating capacity. Thus, the oil and gas generated from the $K_1n_1$ source rock can migrate laterally into $K_1n_2$ after vertical transport along oil-source faults. Through detailed structural interpretations, we determined that this block is controlled by the F4 Fault in the east and forms a fault trap with a structural amplitude of about 15 m (49 ft). In addition, the seismic attribute inversion data confirm that the sand bodies in this trap are well developed and have a certain connectivity (Figure 9b) and the RSF is about 38–46%. The SGR value of the fault for the upward inclination of $K_1n_2$ is 35–56%, and the burial depth is 1320–1710 m (4331–5610 ft). By comparing the actual two factors with the SGR&H threshold of $K_1n_2$ in the Surennuoer area, it can be seen that the actual SGR value of fault rock is greater than the $SGR_{O-MS}$ threshold (33–40%) in most areas, and even greater than the $SGR_{MS-SS}$ threshold (46–52%) in partial areas, so we conclude that the F4 Fault has a moderate to strong sealing ability (Figure 9c). The above accumulation factors correspond

perfectly with each other. Thus, this block is a key block for oil and gas exploration in the Hailar Basin, and an oil bloom can be expected when drilling commences.

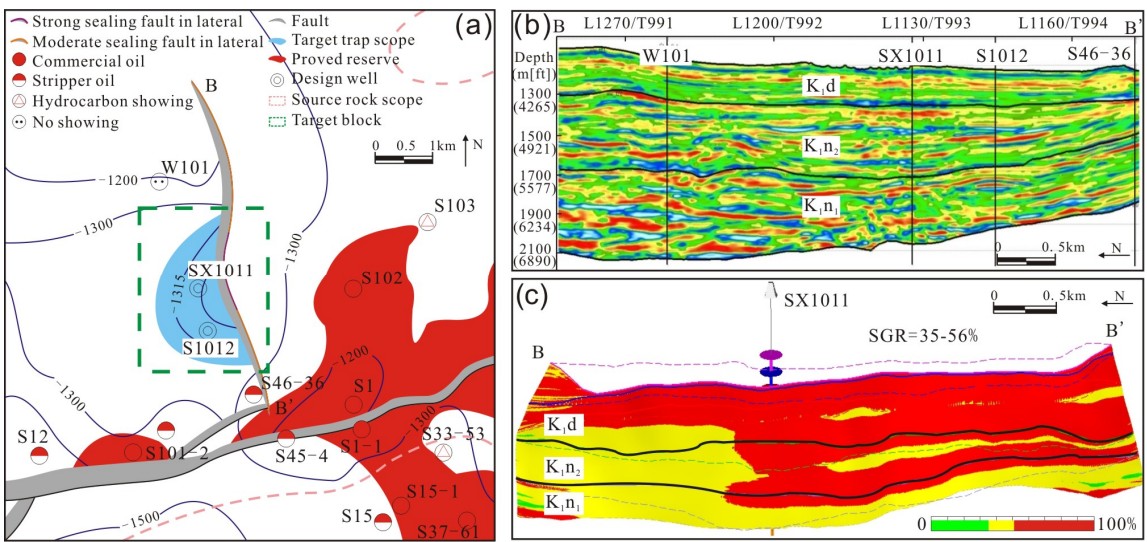

**Figure 9.** Analysis of the target areas for Blocks SX1011-S1012 in the Hailar Basin. (**a**) The location of the target block. (**b**) Inversion profile BB' in Figure 10a. (**c**) SGR attributes of the F4 Fault.

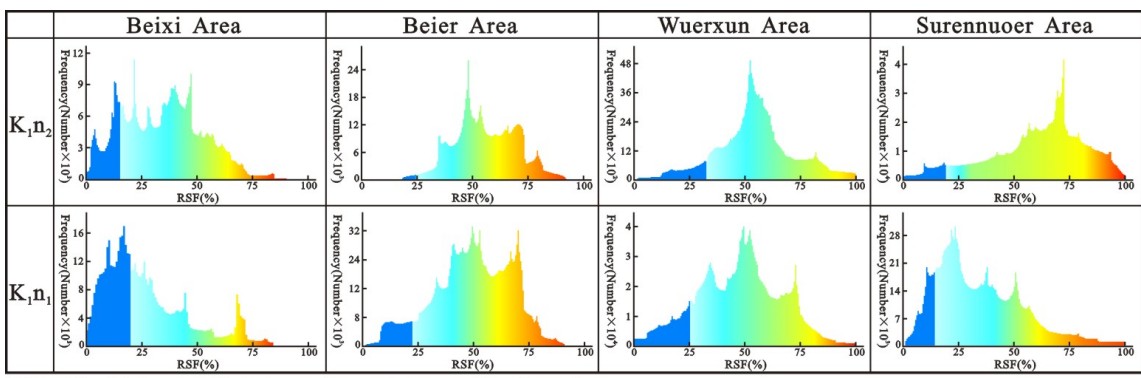

RSF—The thickness ratio between the sandstone and the formation

**Figure 10.** Statistical map of the RSF in the $K_1n$ Formation in different areas of the Hailar Basin.

## 5. Discussion

### 5.1. Differences in the Lateral Sealing Ability of Faults within Different Layers in the Same Area

Based on the above analysis, we conclude that although in general, the Beixi, Beier, Wuerxun, and Surennuoer blocks in the Hailar Basin have similar fault sealing characteristics, this only occurs when the actual SGR value of fault rock is not smaller than the critical $SGR_{O-MS}$ value of the fault lateral sealing at corresponding burial depth. If the fault is laterally sealed, hydrocarbons can accumulate. If the fault is laterally opened, hydrocarbons may leak across the fault. However, by comparing the SGR&H threshold of lateral fault sealing, we conclude that there are some differences in the lateral sealing ability of faults within different layers in the same area and within the same layer in different areas due to the controlling of the fault and formation attributes.

Taking the Surennuoer area of the Wuerxun Depression as an example (Figure 6d), the lateral sealing thresholds of faults within different layers are different. As the burial depth increases from $K_1n_2$ to $K_1n_1$, the $SGR_{O-MS}$ threshold of the sealing gradually decreases from 33–41% to 28–33%. From the logging and seismic date, the depth of the $K_1n_2$ Formation is 1300–1740 m (4265–5709 ft), which indicates that the effective stress on the fault surface due to overlying deposition is relatively small and the degree of diagenesis of the fault rock is

low, so a relatively high SGR value of the fault rock is required to form sealing. While the depth of the $K_1n_1$ Formation is between 1740–2135 m (5709–7005 ft), the diagenesis of the fault rock due to thicker overlying deposits is more obvious, and the degree of diagenesis of the fault rock is high, so a relatively low SGR value can form sealing.

In summary, taking into account the variations in the burial depth of the different layers and the controlling effect of depth on the fault sealing, the deeper the fault rock is buried, the more likely it is to form a lateral seal with comparatively stronger diagenesis over a relatively long period. All these factors cause the threshold of lateral fault sealing to gradually decrease with increasing burial depth.

*5.2. Differences in the Lateral Sealing Ability of Faults within the Same Layer in Different Areas*

The differences in the sealing thresholds are not only reflected within different layers in the same area but also within the same layer in different areas.

From the above analysis, the coupling relationship between the SGR value and the burial depth of fault rock affects the lateral sealing ability of the fault; that is, the deeper the fault is buried, the larger the SGR value of the fault rock, the higher the hydrocarbon column that can be sealed laterally by the fault. However, whether the fault is laterally sealed or open depends on the difference in the leakage capacity between the fault rock and the reservoir rock, namely, the difference in the entry pressure [27,44]. Only when the entry pressure of the fault rock is greater than or equal to that of the reservoir rock can the fault seal hydrocarbons in and form a deposit. At this time, it is meaningful to study the lateral sealing ability of the fault. Therefore, the fault sealing is not only controlled by its own properties, e.g., the SGR value and the degree of diagenesis of the fault rock [72] but also the physical properties of the reservoir. Assuming that the burial depths of faults do not vary much, if the RSF is high, which indicates that the clay content of the reservoir rock and its entry pressure is relatively low, then the minimum entry pressure of the fault rock required to form a seal is correspondingly reduced; that is to say, the higher the RSF, the smaller the SGR&H threshold required for lateral fault sealing to occur.

Based on the above principles, the reasons for the differences in the SGR&H thresholds of fault sealing in the target layers of the different blocks can be discussed in detail. By analyzing the evolution and distribution of the sedimentary facies, as well as the well logging and stratigraphic data, the thickness of sandstone layers and the whole formations in each well are counted, and then the ratio of them are calculated to obtain the RSF values of the $K_1n$ Formation in the different blocks (Figure 10). Taking the $K_1n_1$ Formation as an example, the Surennuoer area is dominated by a fan delta front and shore-shallow lacustrine sediments. Its average RSF is the highest in the entire basin (69%), followed by the Beier and Wuerxun areas. Because the Beixi area locally developed the deep-semi-deep lacustrine sediments, its RSF is only 33%.

By comparing the relationship between the actual SGR value and burial depth of the fault rock, the SGR&H threshold of fault lateral sealing and the layer RSF value, the following can be concluded. (1) For the Surennuoer Area, the $K_1n$ Formation has the shallowest burial depth. With all other influence factors being equal, the SGR&H threshold for fault sealing would be slightly larger than that of other areas in the Hailar Basin according to the method described above, but the actual analysis results reveal (Figure 7) that this SGR&H threshold is significantly smaller than that of other areas. The main reason for this difference is that the RSF value of the $K_1n$ Formation in the Surennuoer Area is relatively high, even though its burial depth is shallow, and the SGR&H threshold required by a lateral sealing mechanism of fault is relatively small. (2) For the Beixi Area, due to the relatively shallow burial depth and relatively low RSF value of the $K_1n$ Formation, its SGR&H threshold is significantly higher than other areas in Hailar basin. (3) For the Beier and Wuerxun Areas, the burial depths of the $K_1n$ Formation are similar, mostly between 2100 m (6890 ft) and 2800 m (9186 ft), and the SGR&H thresholds required for lateral sealing are almost the same: the threshold in Beier Area which has a slightly higher RSF value is slightly lower than that in the Wuerxun Area.

In summary, with decreasing RSF of the target layers, the SGR&H threshold for fault sealing gradually increases, which is consistent with the changes described above. The difference in the reservoir properties is the main reason for the obvious differences in the SGR&H thresholds of lateral fault sealing within the same layer in the Hailar Basin.

*5.3. Advantages and Limitations of the SGR&H Threshold Method*

Through the above analysis, it can be seen that the SGR&H threshold method changes the static SGR threshold value (based only on the clay content of the fault rock previously) into a dynamic SGR threshold value that gradually changes with the burial depth. The latter evaluates the lateral sealing ability of the fault based on more comprehensive factors, and comprehensively analyzes the influence of the changes in internal structure, porosity and permeability of fault rock and the requirement for clay content with the increase in burial depth. Taking the Beier Area as an example, when the SGR threshold method is used for analysis, the threshold value obtained is about 35%. At different depths of all strata, only when the actual SGR value of the fault rock is greater than or equal to 35% can the fault be sealed; otherwise, the fault is open. When using the SGR&H threshold method (Equations (6) and (7), Figure 6b) to evaluate the lateral sealing of the fault, if the buried depth is less than the depth demarcation point, which is about 2250 m, the improved SGR&H threshold is significantly greater than the SGR threshold, that is, the SGR threshold method significantly overestimates the actual sealing ability of the fault. As we all know, faults developed near the earth's surface do not seal easily, due to late cessation of fault activity. If the burial depth is greater than the depth demarcation point (2250 m), the improved SGR&H threshold is significantly smaller than the SGR threshold, which means that the SGR threshold method obviously underestimates the actual sealing ability of the fault.

The evaluation of fault lateral sealing ability by the SGR&H threshold method is carried out under the condition that the activity history of fault is earlier than the hydrocarbon accumulation period, and the SGR value of fault rock used to establish the templates is the clay content of the fault rock in the present period. With the evolution of geological history, the fault rock and surrounding rock are affected by various diageneses, such as compression and cementation. The parameters used to calculate the SGR value of fault rocks (such as formation thickness and fault displacement) are constantly changing, so the current SGR value of fault rocks cannot represent the ancient SGR value at the key moment of hydrocarbon accumulation. There may be a variety of situations, such as the accumulation period and the present both being sealed, the accumulation period and the present are both opened, or the accumulation period is sealed and the present is opened, the accumulation period is opened and the present is sealed. Therefore, in order to more accurately analyze the control of fault lateral sealing on hydrocarbon distribution, it is far from enough to only change the static boundary from space to dynamic boundary, and it is also necessary to carry out dynamic analysis of fault lateral sealing in terms of the time dimension. Therefore, quantitative research on lateral sealing of faults still needs the joint efforts of many scholars.

**6. Conclusions**

(1)  The lateral sealing ability of faults is controlled by multiple geological factors. For the faults developed in the Hailar Basin, which form fault rock seals, the influencing factors are the clay content of the fault rock (SGR), the degree of diagenesis of the fault rock, and the fault activity history. The degree of diagenesis can be represented by the burial depth of the fault rock. With higher SGR of the fault rock, the deeper the burial and the fault activity occurs before hydrocarbon accumulation such that the lateral sealing ability of the fault will be stronger, which is more favorable for oil and gas accumulation.

(2)  The lateral sealing property and the lateral sealing ability of a fault are two different but mutually restrictive concepts. The former depends on the relative difference in the

entry pressures of the fault rock and the reservoir rock, while the latter depends on the relationship between the SGR value and the burial depth of the fault rock. For a set of reservoir rocks, the higher the RSF is, the smaller the critical entry pressure and the lower the SGR&H threshold required for the fault rock to become laterally sealed.

(3)     The lateral sealing ability of faults in different areas and within different layers is different. (a) In the same area, the thresholds of faults sealed in different layers are different because the deep strata are subjected to greater pressures and longer loading, so the faults are more likely to seal laterally; that is to say, the SGR&H threshold is relatively small. (b) Within the same layer, the thresholds for fault sealing in different areas are also different, and the threshold gradually decreases with increasing RSF.

**Author Contributions:** Conceptualization, X.H.; Software, Y.L. (Yang Liu); Investigation, X.H.; Data curation, J.L.; Writing—original draft preparation, X.H.; Writing—review and editing, X.H. and Y.L. (Yangfang Lv). All authors have read and agreed to the published version of the manuscript.

**Funding:** The study is supported by the National Natural Science Foundation of China (No. 42102165, No. 42202158 and No. 42272166) and the Natural Science Foundation of Heilongjiang Province (LH2021D005).

**Data Availability Statement:** The data that support the findings of this study are available on request from the corresponding author upon reasonable request.

**Acknowledgments:** The authors gratefully acknowledge the SGR formula put forward by Graham Yielding and the software of Traptester 7.2 that has been used for fault-plane diagrams.

**Conflicts of Interest:** The authors declare no conflict of interest.

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
