# Peer review of "Quantitative Study of the Lateral Sealing Ability of Faults Considering the Diagenesis Degree of the Fault Rock: An Example from the Nantun Formation in the Wuerxun-Beier Sag in the Hailar Basin, China"

_resources, doi:10.3390/resources12090098_

Round 1
Reviewer 1 Report
Based on previous analyses and summaries of factors affecting the fault closure capacity of clastic strata, the article analyses the lateral closure capacity of faults in terms of the degree of diagenesis of fault rocks, thereby assessing the lateral closure capacity of faults in the Beixi, Bel, Ulson and Sulunor areas of the Hailar Basin, China.
Comments are provided for the authors' information only:
1、The introduction section summarises the methods for evaluating the lateral closure capacity of faults and evaluates the advantages, disadvantages and applicability of the various methods, but does not indicate the advantages and differences between the research methods in this paper compared with those of previous studies.
2、There is little analysis of the results of the use of this assessment method in the case studies and no analysis of the strengths and weaknesses of this assessment method.
3、Figure 9 and Figure 10 are not centred, note the uniform format.
4、The clarity of Figure 10 needs to be improved, as do other places.
5、Note the format of the formulae in the text, some of which are not italicised.
In summary, the essay needs to be revised, paying attention to its coherence and checking the formatting and font of the essay.
Based on previous analyses and summaries of factors affecting the fault closure capacity of clastic strata, the article analyses the lateral closure capacity of faults in terms of the degree of diagenesis of fault rocks, thereby assessing the lateral closure capacity of faults in the Beixi, Bel, Ulson and Sulunor areas of the Hailar Basin, China.
Comments are provided for the authors' information only:
1、The introduction section summarises the methods for evaluating the lateral closure capacity of faults and evaluates the advantages, disadvantages and applicability of the various methods, but does not indicate the advantages and differences between the research methods in this paper compared with those of previous studies.
2、There is little analysis of the results of the use of this assessment method in the case studies and no analysis of the strengths and weaknesses of this assessment method.
3、Figure 9 and Figure 10 are not centred, note the uniform format.
4、The clarity of Figure 10 needs to be improved, as do other places.
5、Note the format of the formulae in the text, some of which are not italicised.
In summary, the essay needs to be revised, paying attention to its coherence and checking the formatting and font of the essay.
Author Response
Dear Editor:
I have made point-by-point response to the reviewer’s comments. The specific comments and modifications are as follows:
Q1:The introduction section summarises the methods for evaluating the lateral closure capacity of faults and evaluates the advantages, disadvantages and applicability of the various methods, but does not indicate the advantages and differences between the research methods in this paper compared with those of previous studies.
A1:Added the advantages of the research methods in this paper compared to previous methods in the introduction section. This method not only quantitatively considers the influence of two parameters, fault clay content and diagenetic degree, on the lateral sealing of faults, but also obtains the configuration relationship between the above two parameters by establishing template.
Q2:There is little analysis of the results of the use of this assessment method in the case studies and no analysis of the strengths and weaknesses of this assessment method.
A2:Section 3.5 of the article mainly expounds the evaluation results obtained by using the new SGR&H method and the advantages of the method compared with the previous methods. Section 3.6 is listed separately to analyzed the typical successful and failed Wells in the study area from the perspective of fault lateral sealing. Section 4 is also a detailed analysis of the differences between lateral opening-medium seal and medium-strong seal of the Beixi, Beier, Wuerxun, and Surennuoer blocks in the Hailar Basin.
Q3:Figure 9 and Figure 10 are not centred, note the uniform format.
A3:All Figures have been adjusted to a uniform format.
Q4:The clarity of Figure 10 needs to be improved, as do other places.
A4:All Figures are processed to make them clearer.
Q5:Note the format of the formulae in the text, some of which are not italicised.
A5:The format of the formula in the text has been checked.
The above questions have been revised in the text.
Due to the limited theoretical level, we don’t know whether the above revises and instructions can meet the reviewers’ requirements, we also look forward to receiving your criticism.
Authors
2023-7-23
Reviewer 2 Report
An interesting article with a high degree of specialization. The authors presented in detail the mechanism of hydrocarbon accumulation in fault zones, taking into account many important factors (e.g. clay content, diagenesis). Clear article with nice figures.
Some explanations of abbreviations are missing, maybe a glossary of abbreviations would be useful?
Author Response
Dear Editor:
I have made point-by-point response to the reviewer’s comments. The specific comments and modifications are as follows:
Q1:Some explanations of abbreviations are missing, maybe a glossary of abbreviations would be useful?
A1:All abbreviations in the article have been checked and the geological significance represented by the abbreviations has been supplemented.
The above questions have been revised in the text.
Due to the limited theoretical level, we don’t know whether the above revises and instructions can meet the reviewers’ requirements, we also look forward to receiving your criticism.
Authors
2023-7-23
Reviewer 3 Report
1- In the abstract, the authors highlighted that this work is based on previous analyses. What are these analyses?
2- "The higher the clay content of the fault rock, the greater the degree of diagenesis, and the stronger the lateral sealing ability of the fault" is it your contribution? if not, this should not be included in the abstract.
3- What is SGR&H? please do not use abbreviation when introducing it for the first time.
4- The abstract is chaotic. The authors should rewrite the abstract with focus on the main findings of the current work.
5- What do you mean by fault-related theories? please rephrase.
6- The English presentation of the introduction requires urgent polishing. The abbreviations should be declared when introduced for the first time.
7- The introduction is not complete. The published works that utilizied the SGR and other methods in similar basins should be disscussed to identify the existing gap in the knowledge.
8- Sections 3.1 and 3.2 are simply review parts and are not suitable for the methodology and analytical technique chapter. These sections should be summarized and included in the introduction.
9- The biggest concern is that the methodology and techniques are mixed with results and disscussions. The authors should urgently reorganize their sections to improve the readability of the manuscript.
The English is poor and must be polished.
Author Response
Dear Editor:
I have made point-by-point response to the reviewer’s comments. The specific comments and modifications are as follows:
Q1:In the abstract, the authors highlighted that this work is based on previous analyses. What are these analyses?
A1:The previous analyses mainly include the analysis of fault lateral sealing in the literature and the understanding obtained by authors through laboratory physical simulation experiments. Among them, for the analysis in the literature, a large number of scholars determined the impact of fault attributes (displacement, clay content, dip angle), stratigraphic attributes (dip angle, sand thickness), their configuration relation , fluid properties, heterogeneity and many other factors on the lateral sealing of faults through cases, analyses and the geological significance contained in different factors may be overlapped. For laboratory physical simulation experiments, the authors have carried out experiments on the effects of dip angel and physical property of fault and that of reservoir on the sealing property, and revealed that the clay content of the fault zone has the highest degree of influence on the sealing property, and its effect is about 3-4 times that of the other two factors.
A large number of exploration, core and experimental analysis have proved that the essence of fault sealing is the capillary pressure difference between fault rock and reservoir rock. Therefore, based on the above, the clay content of fault rock is selected as one of the main controlling factors of method in this article. In addition, from the formation of fault to the present stage, with the increase of burial depth and the enhancement of the grinding degree of fault blocks, the internal structure of the fault zone has significantly differentiated, and the diagenetic pressure of the deep fault is greater than that of the shallow fault and the fault with small dip angle is greater than that of the fault with large dip angle. Therefore, the diagenetic degree of fault rock is chosen as another main factor of method in this article.
Q2:"The higher the clay content of the fault rock, the greater the degree of diagenesis, and the stronger the lateral sealing ability of the fault" is it your contribution? if not, this should not be included in the abstract.
A2:This sentence is summarized through the long-term accumulation of predecessors, rather than the contribution of this article, has been deleted in the abstract.
Q3:What is SGR&H? please do not use abbreviation when introducing it for the first time.
A3:The SGR&H is the configuration relationship between clay content and burial depth of fault rock, the introduction of abbreviations has been supplemented in the corresponding places and the abbreviations in the full text have been checked.
Q4:The abstract is chaotic. The authors should rewrite the abstract with focus on the main findings of the current work.
A4:The abstract section has been reorganized, conclusions not discussed in this article have been deleted, and the SGR&H threshold method and evaluation results of fault lateral sealing ability in the Hailar Basin have been added.
Q5:What do you mean by fault-related theories? please rephrase.
A5:The fault-related theories aims to characterize the fault theory related to the research of fault lateral sealing in this article, such as the transition of fault sealing mechanism from juxtaposition sealing to capillary sealing, the difference between juxtaposition sealing and fault rock sealing, and the development of fault lateral sealing evaluation methods from qualitative to semi-quantitative to quantitative, etc. The actual meaning of the word is corrected.
Q6:The English presentation of the introduction requires urgent polishing. The abbreviations should be declared when introduced for the first time.
A6:The English presentation of the article has been polished. The full geological meaning of the abbreviations have been added.
Q7:The introduction is not complete. The published works that utilized the SGR and other methods in similar basins should be discussed to identify the existing gap in the knowledge.
A7:Corresponding content is added in the introduction. Because the evaluation result of SGR threshold method is a static value, which is the same at different depths. In fact, affected by the diagenesis degree of fault rock, the SGR threshold required at different depths is different, which may lead to the overestimation of the sealing ability of shallow faults by previous methods in evaluating the lateral seal ability of faults, thus drilling into the aqueous layer in the sealing area; However, the sealing ability of deep faults is underestimated, and hydrocarbon are drilled in the opening area. In the past, when analyzing the lateral sealing ability of faults and hydrocarbon distribution in Hailar Basin, this kind of problem can be solved well by using SGR&H threshold method.
Q8:Sections 3.1 and 3.2 are simply review parts and are not suitable for the methodology and analytical technique chapter. These sections should be summarized and included in the introduction.
A8:The simply review parts of Sections 3.1 and 3.2 are adjusted to the introduction, and the methods and techniques in Section 3 are changed into a separate section, which is separate from the evaluation results and discussion on the lateral sealing ability of faults in the Hailar Basin.
Q9:The biggest concern is that the methodology and techniques are mixed with results and discussions. The authors should urgently reorganize their sections to improve the readability of the manuscript.
A9:The content structure of this article has been adjusted, the methods and techniques, results and discussions have been elaborated separately to improve the readability of the manuscript.
The above questions have been revised in the text.
Due to the limited theoretical level, we don’t know whether the above revises and instructions can meet the reviewers’ requirements, we also look forward to receiving your criticism.
Authors
2023-7-23
Reviewer 4 Report
The purpose of this paper is to evaluate the lateral sealing capacity of a fault in a clalolith sequence based on the previous analysis and summary of the factors affecting sealing capacity. Quite an interesting article that has new knowledge. The authors of the article obtained valuable practical results: 1) in the same area, the thresholds for sealing faults within different layers are different, because deep layers are subjected to higher pressures and longer loading times; 2) In the same layer, the thresholds for sealing faults in different areas are also different. In general, the work can be accepted for publication with minimal corrections:
1) Minimum correction of the English language according to the text of the article
2) The list of references must be diluted with newer articles on this topic.
Martyushev, D.A., Chalova, P.O., Davoodi, S., Ashraf, U. Evaluation of facies heterogeneity in reef carbonate reservoirs: A case study from the oil field, Perm Krai, Central-Eastern Russia. Geoenergy Science and Engineering. 2023. 227. 211814. https://doi.org/10.1016/j.geoen.2023.211814
Makarian, E., Abad, A.B.M.N., Manaman, N.S., Mansourian, D., Elyasi, A., Namazifard, P., Martyushev, D. An efficient and comprehensive poroelastic analysis of hydrocarbon systems using multiple data sets through laboratory tests and geophysical logs: a case study in an iranian hydrocarbon reservoir. Carbonates and Evaporites. 2023. 38. 37. https://doi.org/10.1007/s13146-023-00861-1
Martyushev, D.A., Ponomareva, I.N., Chukhlov, A.S., Davoodi, S., Osovetsky, B.M., Kazymov, K.P., Yang, Y. Study of void space structure and its influence on carbonate reservoir properties: X-ray microtomography, electron microscopy, and well testing. Marine and Petroleum Geology. 2023. 151. 106192. https://doi.org/10.1016/j.marpetgeo.2023.106192
In general, there are no questions in English. In some places of the text it is necessary to check the grammar and punctuation, and correct a number of errors.
Author Response
Dear Editor:
I have made point-by-point response to the reviewer’s comments. The specific comments and modifications are as follows:
Q1:Minimum correction of the English language according to the text of the article.
A1:We have read the entire article and made modifications and improvements to the English language.
Q2:The list of references must be diluted with newer articles on this topic.
Martyushev, D.A., Chalova, P.O., Davoodi, S., Ashraf, U. Evaluation of facies heterogeneity in reef carbonate reservoirs: A case study from the oil field, Perm Krai, Central-Eastern Russia. Geoenergy Science and Engineering. 2023. 227. 211814. https://doi.org/10.1016/j.geoen.2023. 211814
Makarian, E., Abad, A.B.M.N., Manaman, N.S., Mansourian, D., Elyasi, A., Namazifard, P., Martyushev, D. An efficient and comprehensive poroelastic analysis of hydrocarbon systems using multiple data sets through laboratory tests and geophysical logs: a case study in an iranian hydrocarbon reservoir. Carbonates and Evaporites. 2023. 38. 37. https://doi.org/10.1007/ s13146-023-00861-1
Martyushev, D.A., Ponomareva, I.N., Chukhlov, A.S., Davoodi, S., Osovetsky, B.M., Kazymov, K.P., Yang, Y. Study of void space structure and its influence on carbonate reservoir properties: X-ray microtomography, electron microscopy, and well testing. Marine and Petroleum Geology. 2023. 151. 106192. https://doi.org/10.1016/j.marpetgeo.2023.106192
A2:The articles provided by reviewer are mainly for carbonate rock formations, and the study area in this paper is dominated by clastic rock formations. Therefore, six articles in recent years are cited for the research topic on the lateral sealing of faults in clastic rock formations and the geological situation of the study area.
The above questions have been revised in the text.
Due to the limited theoretical level, we don’t know whether the above revises and instructions can meet the reviewers’ requirements, we also look forward to receiving your criticism.
Authors
2023-7-23
Reviewer 5 Report
Review for Hu et al. “Quantitative Study of the Lateral Sealing Ability of FaultsConsidering the Diagenesis Degree of the Fault Rock: An Example from the Nantun
Formation in the Wuerxun-Beier Sag in the Hailar Basin, China”, submitted to
Resources.
This is an interesting manuscript on fault seal analysis. The manuscript is suitable for a wide audience in the petroleum geoscience community. However, I have listed a few points that should be considered prior to publication. I also uploaded an annotated version of the manuscript in which I indicated these issues.
Abstract
Page 1, Line 1-2: What is “clasolite stratum”? Please specify.
Introduction
Page 1, Line 3: What sedimentary characteristics do you mean? Please specify.
Page 1, Line 5-7: Please rephrase this sentence. I do not understand the meaning.
Page 1, Line 10: What do you mean with “fault-related theories”? Please specify.
Page 2, Line 1-2: Please specify the “fault attributes”. Do you mean displacement, length and thickness?
Geological setting
Page 3, Fig. 1: Could you add coordinates to the map?
Page 4, Fig. 2: Maybe you could add a column with the stages.
Page 4, Line 2-3: What is a “residual basin”? and what do you mean with “Fault-depression transformation”? Please specify and use the common rift basin terminology.
In the chapter “3.1 Fault zone architecture”, key literature should be added. At the end of the first sentence “A fault zones consists of two architectural elements……” please cite the following literature:
Chester, F.M., Logan, J.M., 1986. Implications for mechanical properties of brittle faults from observations of the Punchbowl fault zone, California. Pure and Applied Geophysics 124, 79-106.
Childs, C., Manzocchi, T., Walsh, J.J., Bonson, C.G., Nicol, A., Schöpfer, M.P.J., 2009. A geometric model of fault zone and fault rock thickness variations. Journal of Structural Geology 31, 117-127.
Faulkner, D.R., Mitchell, T.M., Jensen, E., Cembrano, J., 2011. Scaling of fault damage zones with displacement and the implications for fault growth processes. Journal of Geophysical Research 116, B05403, doi:10.1029/2010JB007788.
Brandes, C & Tanner, D. (2020) Fault mechanics and earthquakes. In: Tanner, D. & Brandes, C. (eds) Understanding Faults: Detecting, Dating, and Modeling, Elsevier, 11-80.
At the end of the second sentence “…which represent intense deformation…” please also cite:
Sibson, R.H., 1977. Fault rocks and fault mechanisms. Journal of the Geological Society of London 133, 191-213.
Shipton, Z.K., Soden, A.M., Kirkpatrick, J.D., Bright, A.M., Lunn, R.L., 2006. How thick is a fault? Fault displacement-thickness scaling revisited. Earthquakes: Radiated energy and the physics of faulting. Geophysical Monograph Series 170, 193-198.
In the next sentence you cite Tanaka et al., (2001), please also cite:
Choi, J.-H., Edwards, P., Ko., K., Kim., Y.-S., 2016. Definition and classification of fault damage zones: A review and new methodological approach. Earth-Science Reviews 152, 70-87.
At the end of this paragraph “…basic characteristics of the surrounding rocks (Bruhn et al., 1994….” please also cite:
Vermilye, J.M., Scholz, C.H., 1998. The process zone: a microstructural view of fault growth. Journal of Geophysical Research 103, B6, 12223-12237.
In chapter “3.2 Type of lateral fault sealing” you should cite the two most important references on fault rocks:
Killick, A.M., 2003. Fault rock classification: an aid to structural interpretation in mine and exploration geology. South African Journal of Geology 106, 395–402.
Woodcock, N.H., Mort, K., 2008. Classification of fault breccias and related fault rocks. Geological Magazine 145, 435-440.
At the end of page 5, “…the thicker the fault zone thickness…” you should cite:
Evans, J.P., 1990. Thickness-displacement relationships for fault zones, Journal of Structural Geology, 12, 1061-1065,
Page 7, Fig. 4: Could you add a depth scale in meter?
Page 7, first paragraph: Does this text describe the result of the current study or from previous studies? You should probably cite some literature here.
Page 8: Is the SGR algorithm really an algorithm or just a formula?
Page 13, Fig. 7: This figure is to small and it is difficult to read the letters and the details in the lithological log. Please enlarge.
Page 14: Fig. 8: What are the red and blue line on the seismic section? Please explain.
Page 16, last paragraph: What is a “fault screened trap”? Do you mean a “fault trap”?
A discussion is completely missing. From my point of view, a discussion is the main part of a scientific paper. In a discussion, I expect a reflection of the data and the results in the view of previous research, or in the view of comparable methods. A discussion could be a comparison of your results with other studies from the literature. There are many studies on the sealing effects of faults. What is the advantage of your approach? In the discussion, the advantages and limitations of the data and of the approach used in this study, as well as the limitations of the interpretation of the results should be clearly addressed.

Author Response
Dear Editor:
I have made point-by-point response to the reviewer’s comments. The specific comments and modifications are as follows:
Q1:Abstract
Page 1, Line 1-2: What is “clasolite stratum”? Please specify.
A1:It's actually siliciclastic and the content has been revised
Q2:Introduction
Page 1, Line 3: What sedimentary characteristics do you mean? Please specify.
Page 1, Line 5-7: Please rephrase this sentence. I do not understand the meaning.
Page 1, Line 10: What do you mean with “fault-related theories”? Please specify.
Page 2, Line 1-2: Please specify the “fault attributes”. Do you mean displacement, length and thickness?
A2:(1)The sedimentary characteristics mainly refer to the time-space distribution characteristics of sedimentary facies, the terms of which were revised in the article. Since the fillings in the fault zone is formed by debris falling into it during the activity history of fault , the distribution characteristics of sedimentary facies in time and space control the change of lithology, which leads to the difference of fault sealing.
(2)Reorganize the language and modify the expression of the statement.
(3)The fault-related theories aims to characterize the fault theory related to the research of fault lateral sealing in this article, such as the transition of fault sealing mechanism from juxtaposition sealing to capillary sealing, the difference between juxtaposition sealing and fault rock sealing, and the development of fault lateral sealing evaluation methods from qualitative to semi-quantitative to quantitative, etc. The actual meaning of the word is corrected.
(4)It is described in detail in the manuscript, mainly including the thickness and clay content of strata which surrounding the fault throw interval as well as the displacement of fault.
Q3:Geological setting
Page 3, Fig. 1: Could you add coordinates to the map?
Page 4, Fig. 2: Maybe you could add a column with the stages.
Page 4, Line 2-3: What is a “residual basin”? and what do you mean with “Fault-depression transformation”? Please specify and use the common rift basin terminology.
A3:(1)Added the coordinates in Figure 1.
(2)Added the tectonic stages of the Hailar Basin in Figure 2, including the rift stage and the rift pahse.
(3)Modified the corresponding language.
Q4:In the chapter “3.1 Fault zone architecture”, key literature should be added. At the end of the first sentence “A fault zones consists of two architectural elements……” please cite the following literature:
Chester, F.M., Logan, J.M., 1986. Implications for mechanical properties of brittle faults from observations of the Punchbowl fault zone, California. Pure and Applied Geophysics 124, 79-106.
Childs, C., Manzocchi, T., Walsh, J.J., Bonson, C.G., Nicol, A., Schöpfer, M.P.J., 2009. A geometric model of fault zone and fault rock thickness variations. Journal of Structural Geology 31, 117-127.
Faulkner, D.R., Mitchell, T.M., Jensen, E., Cembrano, J., 2011. Scaling of fault damage zones with displacement and the implications for fault growth processes. Journal of Geophysical Research 116, B05403, doi:10.1029/2010JB007788.
Brandes, C & Tanner, D. (2020) Fault mechanics and earthquakes. In: Tanner, D. & Brandes, C. (eds) Understanding Faults: Detecting, Dating, and Modeling, Elsevier, 11-80.
At the end of the second sentence “…which represent intense deformation…” please also cite:
Sibson, R.H., 1977. Fault rocks and fault mechanisms. Journal of the Geological Society of London 133, 191-213.
Shipton, Z.K., Soden, A.M., Kirkpatrick, J.D., Bright, A.M., Lunn, R.L., 2006. How thick is a fault? Fault displacement-thickness scaling revisited. Earthquakes: Radiated energy and the physics of faulting. Geophysical Monograph Series 170, 193-198.
In the next sentence you cite Tanaka et al., (2001), please also cite:
Choi, J.-H., Edwards, P., Ko., K., Kim., Y.-S., 2016. Definition and classification of fault damage zones: A review and new methodological approach. Earth-Science Reviews 152, 70-87.
At the end of this paragraph “…basic characteristics of the surrounding rocks (Bruhn et al., 1994….” please also cite:
Vermilye, J.M., Scholz, C.H., 1998. The process zone: a microstructural view of fault growth. Journal of Geophysical Research 103, B6, 12223-12237.
In chapter “3.2 Type of lateral fault sealing” you should cite the two most important references on fault rocks:
Killick, A.M., 2003. Fault rock classification: an aid to structural interpretation in mine and exploration geology. South African Journal of Geology 106, 395–402.
Woodcock, N.H., Mort, K., 2008. Classification of fault breccias and related fault rocks. Geological Magazine 145, 435-440.
At the end of page 5, “…the thicker the fault zone thickness…” you should cite:
Evans, J.P., 1990. Thickness-displacement relationships for fault zones, Journal of Structural Geology, 12, 1061-1065.
A5:Added to the above references in the appropriate space.
Q5:Page 7, Fig. 4: Could you add a depth scale in meter?
Page 7, first paragraph: Does this text describe the result of the current study or from previous studies? You should probably cite some literature here.
Page 8: Is the SGR algorithm really an algorithm or just a formula?
Page 13, Fig. 7: This figure is to small and it is difficult to read the letters and the details in the lithological log. Please enlarge.
Page 14: Fig. 8: What are the red and blue line on the seismic section? Please explain.
Page 16, last paragraph: What is a “fault screened trap”? Do you mean a “fault trap”?
A5:(1)Figure 4 (changed to Figure 3 after reorganized the article section) is only a schematic diagram, indicating that there is such a trend in underground that the diagenesis degree increases and the permeability deteriorates with the increase of burial depth, but the change characteristics are different in different regions. For example, the depth of cementation is also related to the geothermal gradient and subsurface hydrothermal.
(2)This paragraph is the result from previous studies to illustrate the influence of the activity history of fault on fault sealing of Hailar Basin. On the basis of the original cites, several references are added.
(3)The SGR algorithm is actually a calculation using the SGR formula, and I feel that it is more appropriate to adjust to the SGR formula, so the content of is modified.
(4)The font size in Figure 7 has been enlarged somewhat.
(5)The meaning of the lines is supplemented in Figure 8b, where the yellow line on the seismic section is the top interface of K1d1, and the red, blue, and pink lines are represented the top interface of K1n2, K1n1 and K1t respectively.
(6)Yes, it means the “fault trap”, modifications have been made in the text.
Q6:A discussion is completely missing. From my point of view, a discussion is the main part of a scientific paper. In a discussion, I expect a reflection of the data and the results in the view of previous research, or in the view of comparable methods. A discussion could be a comparison of your results with other studies from the literature. There are many studies on the sealing effects of faults. What is the advantage of your approach? In the discussion, the advantages and limitations of the data and of the approach used in this study, as well as the limitations of the interpretation of the results should be clearly addressed.
A6:The discussion section is supplemented to analyze the evaluation results of this article and the advantages and limitations of the established method.
The above questions have been revised in the text.
Due to the limited theoretical level, we don’t know whether the above revises and instructions can meet the reviewers’ requirements, we also look forward to receiving your criticism.
Authors
2023-7-23
Round 2
Reviewer 3 Report
1- Line 10: rephrase to "Content of clay mineral phase and the diagenetic degree of fault rock were investigated as the main factors to evaluate the lateral sealing of faults".
2- Line 112: What it refers to? please rephrase.
3- Line 115: "This method" which method you mean?
4- Figure 1: I think the structural patterns added in the figure is not the own work of the authors. Please add relevant references.
5- Figure 2: see the comment above.
The English is fine except with a few edits I have mentioned in the comments
Author Response
Dear Editor:
I have made point-by-point response to the reviewer’s comments. The specific comments and modifications are as follows:
Reviewer 3:
Q1:Line 10: rephrase to "Content of clay mineral phase and the diagenetic degree of fault rock were investigated as the main factors to evaluate the lateral sealing of faults".
A1:The sentence has been rephrased according to the opinions of the reviewer.
Q2:Line 112: What it refers to? please rephrase.
A2:"It" refers to the SGR&H threshold method established in this article, which has been rephrased in the corresponding parts.
Q3:Line 115: "This method" which method you mean?
A3:"This method" refers to the SGR&H threshold method, , which has been rephrased in the corresponding parts.
Q4:Figure 1: I think the structural patterns added in the figure is not the own work of the authors. Please add relevant references.
A4:Relevant references to structural patterns in Figure 1 have been added, and the remaining parts of the figure were drawn by the authors according to the characteristics of fault in the 3D seismic interpretation and the actual distribution of oil, gas and aqueous. In addition, there is no copyright issue with this figure.
Q5:Figure 2: see the comment above.
A5:Relevant references to tectonic stages in Figure 2 have been added, and the remaining parts of the figure were drawn by the authors according to the lithological characteristics of the actual well and the combination of the source, reservoir, seal rock.
The above questions have been revised and marked in blue in the manuscript.
Due to the limited theoretical level, we don’t know whether the above revises and instructions can meet the reviewers’ requirements, we also look forward to receiving your criticism.
Authors
2023-8-7